# Synthesis of 4′-Substituted Carbocyclic Uracil Derivatives and Their Monophosphate Prodrugs as Potential Antiviral Agents

**DOI:** 10.3390/v15020544

**Published:** 2023-02-16

**Authors:** Nicolas G. Biteau, Sarah A. Amichai, Niloufar Azadi, Ramyani De, Jessica Downs-Bowen, Julia C. Lecher, Tamara MacBrayer, Raymond F. Schinazi, Franck Amblard

**Affiliations:** Laboratory of Biochemical Pharmacology, Center for ViroScience and Cure, Department of Pediatrics, Emory University School of Medicine, and Children’s Healthcare of Atlanta, Atlanta, GA 30322, USA

**Keywords:** antivirals, prodrugs, carbocyclic nucleosides, viral polymerase, SARS-CoV-2, influenza virus, norovirus

## Abstract

Over the past decades, both 4′-modified nucleoside and carbocyclic nucleoside analogs have been under the spotlight as several compounds from either family showed anti-HIV, HCV, RSV or SARS-CoV-2 activity. Herein, we designed compounds combining these two features and report the synthesis of a series of novel 4′-substituted carbocyclic uracil derivatives along with their corresponding monophosphate prodrugs. These compounds were successfully prepared in 19 to 22 steps from the commercially available (-)-Vince lactam and were evaluated against a panel of RNA viruses including SARS-CoV-2, influenza A/B viruses and norovirus.

## 1. Introduction

Nucleoside analogs targeting viral polymerases are the cornerstone of current antiviral therapies with numerous compounds approved for the treatment of infectious diseases such as human immunodeficiency virus (HIV), hepatitis C virus (HCV), hepatitis B virus (HBV), herpes simplex virus (HSV) or SARS-CoV-2 [1,2]. The chemistry around each position of the sugar moiety of natural ribo- or 2′-deoxyribonucleosides has been explored extensively, but more recently, synthetically more challenging 4′-modifications have received a great deal of attention [3]. For instance, 4′-azidocytidine or its methyl ester prodrug Balapiravir (**1**) along with its 4′-cyano analogs (**2**) [4], have been reported in their 5′-triphosphate form as HCV polymerase inhibitors [5]. Additionally, 4′-chloromethyl-2′-deoxy-2′-fluoro-cytidine (**3**) (ALS-8812) as its *iso*propyl ester prodrug (ALS-8176) (**4**) were, until recently, studied in a phase II clinical trial as a new treatment for respiratory syncytial virus (RSV) [6,7].

4′-Ethynyl-2′-deoxy-2-fluoro-adenosine (**5**) (EFdA/MK8591/Islatravir), the most potent in vitro anti-HIV nucleoside analog inhibitor known to date [8], is being clinically evaluated as a subdermal implant for HIV treatment and prophylaxis [9,10,11]. More recently, 4′-fluoro-uridine (**6**) has been reported as a potent inhibitor of RSV and SARS-CoV-2 replication [12,13] (Figure 1). Even though a number of 4′-modified ribo- or deoxyribo nucleosides have been synthesized, their carbanucleoside versions, in which the oxygen (O) in the sugar ring has been replaced by a carbon (CH_2_), represent an understudied family of nucleoside analogs. Several examples [14,15,16], including approved drugs entecavir and abacavir, show that this type of modification does not impact the cellular processing of the nucleoside while making compounds more resistant to nucleoside phosphorylase responsible of the *N*-glycosidic bond cleavage [17]. Indeed, 4′-methyl analogs (**7**) [18] were shown to display weak activities against a panel of RNA viruses including yellow fever (YF), dengue virus (DENV), venezuelan equine encephalis virus (VEE) and west nile virus (WNV), while 4′-ethynyl and 4′-cyano carbocyclic-2′-deoxyribonucleoside analogs (**8a**) and (**8b**) (Figure 2) displayed anti-HIV-1 activity [19] (Figure 2).

Based on these precedents, we wish to report herein the synthesis of novel 4′-substituted carbocyclic uridine analogs (**9**–**11**) and their corresponding monophosphate prodrugs (**12**–**14**) and report their evaluation against a small panel of RNA viruses including Norovirus, Influenza A, Influenza B viruses and SARS-CoV-2 (Figure 3).

## 2. Material and Methods

### 2.1. Antiviral Assays

#### 2.1.1. SARS-CoV-2 Antiviral Assays

The anti-SARS-CoV-2 activity of compounds herein prepared was evaluated at 10 μM following previously reported methods [20]. Briefly, a monolayer of Vero cells in a 96-well cell culture microplate was treated with 10 μM of each compound for 1 h followed by infection with SARS-CoV-2 (Washington Strain) at 0.1 MOI. After 1 h adsorption at 37 °C, the virus inoculum was removed, and the compound or vehicle-containing medium was added to the respected wells. Resultant virus progeny yield was measured 2 days post-treatment from the supernatant of treated infected cells by specific quantitative RT-PCR.

#### 2.1.2. Norovirus Antiviral Assays

The anti-NoV activity of compounds herein prepared was evaluated at 10 μM following previously reported methods [21,22]. Briefly, HG23 replicon cells, kindly provided by Kyeong-Ok Chang, Kansas State University (Manhattan, KS, USA), were seeded at a density of 1.6 × 10^4^ cells/well in 96-well plates and incubated at 37 °C and 5% CO_2_ overnight. Compounds were tested at 10 µM. Compounds were added in triplicate to 80–90% confluent monolayers and incubated at 37 °C and 5% CO_2_. Untreated cells were incubated in each plate. At 24, 48, 72 and 96 h post-treatment, total RNA was extracted using the Mag-Max Total RNA Isolation kit (Ambion, Austin, TX, USA) and NV replicon RNA was quantified by GI NoV Taqman real-time RT-PCR (NoV RT-qPCR). Protein expression levels were monitored by western blot analysis.

#### 2.1.3. Influenza A/B Antiviral Assays

The anti- Influenza A/B activity of compounds herein prepared was evaluated at 40 μM following previously reported methods [23]. Briefly, A549 cells were seeded at a density of 35,000 cells/well and incubated at 37 °C and 5% CO_2_ overnight. Cells were treated with test compound at 40 µM, then incubated at 37 °C and 5% CO_2_ for 1 h before being inoculated with 0.1 MOI (PR8-PB2-Gluc) or 1.0 MOI (Ya88-PB1-NanoLuc).

### 2.2. Cytotoxicity Assays

The cytotoxicity of the compounds was evaluated using previously reported methods assays [24]. Briefly, in vitro cytotoxicity was determined using the CellTiter 96 non-radioactive cell proliferation colorimetric assay (MTT assay, Promega, Madison, WI, USA) in primary human peripheral blood mononuclear (PBM), human T lymphoblast (CEM), human hepatocellular carcinoma (Huh7) and kidney epithelial (Vero) cell lines. Toxicity levels were measured as the concentration of test compound that inhibited cell proliferation by 50% (CC_50_).

## 3. Results

Compounds (**9**–**11**) along with their corresponding monophosphate prodrugs (**12**–**14**) were successfully synthesized, characterized chemically, and evaluated for antiviral activity and cytotoxicity. Their toxicity profile was assessed and none of them displayed toxicities up to 100 μM in primary human PBM cells, CEM cells, Vero cells and human liver cells (Huh7). Furthermore, none of them exhibited anti- SARS-CoV-2, Influenza A/B viruses and Norovirus. activities at concentration up to 10 μM. Appropriate positive controls exhibited significant activity (SARS-CoV-2: Remdesivir; Influenza A/B viruses: baloxavir; Norovirus: 2’-C-methylcytidine).

## 4. Discussion

The nucleosides (**9**–**11**) and their corresponding monophosphate prodrugs (**12**–**14**) were synthesized by following the chemistry described in Figure 1, Figure 2 and Figure 3 (see Appendix A for detailed protocols and full characterizations).

The targeted 4′-alkyno, 4′-cyano and 4′-chloromethyl carbocyclic ribonucleoside analogs (**9**–**11**) along with their corresponding prodrugs (**12**–**14**) were prepared from the same carbocyclic uridine precursor (**15**), which was synthetized from the (-)-Vince lactam in 8 steps (47% yield) following reported procedures [25,26,27]. Compound (**15**) was first *per*-silylated with *tert*-butyldimethylsilyl chloride (TBDMSCl) in presence of imidazole before selective 5′-deprotection, in presence of trifluoroacetic acid (TFA) and water, to afford compound (**16**) in 62% yield over 2 steps. Compound (**16**) was then oxidized with Dess–Martin periodinane (DMP) to form the corresponding aldehyde intermediate followed by further reaction with 37% in water paraformaldehyde in presence of sodium hydroxide and final reduction with NaBH_4_ to afford 4′-diol intermediate (**17**) in 70% yield over 3 steps. Compound (**20**) was synthesized from (**17**) through a 3-step process by selective protection of the 4′-α-hydroxymethyl group with 4,4’-dimethoxytrityl chloride (DMTrCl), followed by protection of the 4′-β-hydroxymethyl group with TBDMSCl, and then selective 5′-DMTr deprotection under acidic conditions (32% yield over 3 steps) (Figure 1).

It is worth noting that the selectivity of this protection/deprotection sequence was later confirmed indirectly via NOESY ^1^H-NMR experiments of the final 4′-ethynyl carbanucleoside (**9**) (Figure 4). Thus, key NOE effects were observed with one of the H^6′^ which couples with H^2′^, H^3′^ and one of the H^5′^, validating the 4′-α-orientation of the alkyne group in compound (**9**).

The desired 4′-alkyno, 4′-cyano and 4′-chloromethyl carbonucleoside analogs (**9**–**11**) were prepared from compound (**20**) according to the chemistry reported in Figure 2. Intermediate (**20**) was first oxidized with DMP to form the corresponding 5′-aldehyde intermediate (**21**) before being engaged in a Seyferth-Gilbert homologation in presence of dimethyl (1-diazo-2-oxopropyl)phosphonate (reagent A) and K_2_CO_3_ to give intermediate (**22**). After 2′-,3′-TBDMS deprotection in the presence of tetra-*n*-butylammonium fluoride (TBAF), the 4′-ethynyl carbocyclic uridine (**9**) was obtained in 49% yield over 3 steps. From the same 5′-aldehyde intermediate (**21**), reaction with hydroxylamine hydrochloride in pyridine followed by dehydration of intermediate (**23**) in presence of the Burgess reagent and final deprotection with TBAF afforded compound (**10**) in 44% in yield over 4 steps. Finally, 4′-chloromethyl uridine analog (**11**) was obtained from (**20**) by first, microwave-activated chlorination with PPh_3_ and CCl_4_ in DCE to afford compound (**24**) in 67% in yield and further deprotection using TBAF.

In order to express their therapeutic effect, nucleoside analogs must be phosphorylated to their corresponding 5′-triphosphate forms by three different kinases. Interestingly, the first phosphorylation is often the limiting step in this process and several monophosphate prodrugs that can be cleaved intracellularly to deliver the monophosphate form of a nucleoside have been developed [28]. Among them, phosphoramidate prodrugs (Protides) as seen in approved drugs such as sofosbuvir (HCV), tenofovir alafenamide (HIV), or remdesivir (SARS-CoV-2) have been well studied and their use has been validated clinically. Thus, the corresponding monophosphate prodrug of nucleosides (**12**–**14**) were prepared through the chemistry described in Figure 3. Compound (**9**) was first protected as a 2′-,3′-acetonide intermediate (**25**) in 64% yield, before reacting it with isopropyl ((*S*)-(perfluorophenoxy)-(phenoxy)phosphoryl)-*L*-alaninate [29] (Reagent B) in presence of *t*-BuMgCl. Final deprotection under acidic conditions gave prodrug (**12**) in 34% yield over 2 steps. On the other hand, prodrugs (**13**) and (**14**) were prepared in 13% and 5% yields respectively, directly from the corresponding parent nucleosides (**10**) and (**11**) by reaction with Reagent B in presence of *t*-BuMgCl at 0 °C overnight. It is worth noting that, the protection of the 2′-3′-positions in compounds (**10**) and (**11**) (as described above for the synthesis of prodrug (**12**)) to form the corresponding prodrugs did not improve the overall yield as both 2′,3′-isopropylidene monophosphate prodrugs appeared to be unstable under deprotection conditions.

## 5. Conclusions

A series of novel 4′-substituted carbocyclic uracil derivatives containing 4′-ethynyl, 4′-cyano and 4′-chloromethyl groups were synthetized in 19 to 22 steps from the (-)-Vince lactam and evaluated against a small panel of clinically relevant RNA viruses. Unfortunately, none of these compounds, nor their corresponding monophosphate prodrugs, displayed significant antiviral activity against SARS-CoV-2, IFV-A, IFV-B or norovirus. It is worth noting though that none of them showed toxicities up to 100 μM in a panel of cell lines including PBM, CEM, Vero and Huh7 cells.

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
