# Peer review of "Synthesis of 4′-Substituted Carbocyclic Uracil Derivatives and Their Monophosphate Prodrugs as Potential Antiviral Agents"

_viruses, 2023, doi:10.3390/v15020544_

Round 1
Reviewer 1 Report
This paper describes the synthesis of three 4'-substituted carbocyclic uridine analogs and their corresponding monophosphate prodrogues, as well as their evaluation against a small panel of RNA viruses.
Compounds 9-11 were obtained from (-)-Vince lactam via a key carbocyclic uridine precursor according to reported procedures. The synthesis of these carbocyclic derivatives is tedious (19-22 steps). Modest to low yields (5-34%) were obtained for the preparation of prodrugs, while good yields are generally reported (see ref 28). The authors should comment about this (in the context of 4'-substituted ribonucleosides).
Unfortunatly none of these 6 compounds showed antiviral activities.
The manuscript is well written, the experimental procedures are clearly described, and all synthetic intermediates and products are fully characterized.
Despite the absence of biological activities, this work provides valuable contributions about this class of nucleoside analogs. In my opinion, this work is acceptable for publication after minor revision.
Minor corrections
(a) As the synthesized compounds are focussed on (limited to) uridine analogs, title, abstract and conclusions should be modified accordingly.
(b) page 4, lane 102 : insert references for compound 15 (not commercially available)
(c) Are ref 25,26 the good ones or in the right place ?
... prepared from the same carbocyclic uridine precursor (15), which was synthetized from the (-)-Vince lactam in 8 steps (47% yield) following reported procedures [25,26].
Compound 15 was described for example in Journal of the Chemical Society. Perkin transactions I, 1986, p. 399 - 404 or in a more recent paper (and cited ref) from Manoharan's lab in Org. Lett. 2022, 24, 2, 525–530. These works should be added to the ref list.
Author Response
- “As the synthesized compounds are focused on (limited to) uridine analogs, title, abstract and conclusions should be modified accordingly.”
In the title, abstract and conclusion, the terms “4’-substituted carbocyclic ribonucleosides” have been substituted by “4’-Substituted Carbocyclic Uracil Derivatives”
- “page 4, lane 102 : insert references for compound 15 (not commercially available)”
The reference for compound 15 has been inserted (It is worth noting though that, at Reviewer2’s request this paragraph was moved into the SI.
- “Are ref 25,26 the good ones or in the right place ? “... prepared from the same carbocyclic uridine precursor (15), which was synthetized from the (-)-Vince lactam in 8 steps (47% yield) following reported procedures [25,26].”
Yes
- “Compound 15 was described for example in Journal of the Chemical Society. Perkin transactions I, 1986, p. 399 - 404 or in a more recent paper (and cited ref) from Manoharan's lab in Org. Lett. 2022, 24, 2, 525–530. These works should be added to the ref list.”
Journal of the Chemical Society. Perkin transactions I, 1986, p. 399 – 404 describes the synthesis of compound 15 through a different route – That is why the reference was not included.
On the other hand, the Org. Lett. 2022, 24, 2, 525–530 reference was included
Reviewer 2 Report
The manuscript, titled “Synthesis of 4'-Substituted Carbocyclic Ribonucleoside Analogs and their Monophosphate Prodrugs as Potential Antiviral Agents” by Dr. Amblard, et al., describes the synthesis of 4'-substituted carbocyclic uridine analogues and their monophosphate “potential” prodrugs.
Here are my comments:
1) The title says ribonucleoside analogues, whereas the manuscript only covers uracil and not the other nucleoside bases. I would suggest the title to reflect that.
2) While it is coming from a reputed group in the field and has Dr. Schinazi’s name as one of the authors, it fails to live up to the novelty part the authors claimed. The chemistry is not new, the compounds do not have a lot of structural diversity, there are no SAR or computational studies. There are overlaps in structures or chemistry with other publications and patents the authors themselves cited.
3) Moreover, right before and after Sofosbuvir, this particular monophosphate prodrug approach has been well-studied by researchers from Idenix, Gilead, Pharmasset, Merck, etc. and numerous other academic researchers.
4) But, it does serve one purpose: these compounds were not active against the viral strains mentioned, so other researchers need not repeat them.
So, I would recommend the authors to add the synthesis and characterization details as a supporting information file and submit this study as a letter or update; either to this issue/journal if the editor approves it or to another journal.
On a side note: The authors mentioned a supporting information file, but I did not find it. Is this a technical error at MDPI?
Author Response
- “The title says ribonucleoside analogues, whereas the manuscript only covers uracil and not the other nucleoside bases. I would suggest the title to reflect that.”
In the title, abstract and conclusion, the terms “4’-substituted carbocyclic ribonucleosides” have been substituted by “4’-Substituted Carbocyclic Uracil Derivatives”
- “While it is coming from a reputed group in the field and has Dr. Schinazi’s name as one of the authors, it fails to live up to the novelty part the authors claimed. The chemistry is not new, the compounds do not have a lot of structural diversity, there are no SAR or computational studies. There are overlaps in structures or chemistry with other publications and patents the authors themselves cited. Moreover, right before and after Sofosbuvir, this particular monophosphate prodrug approach has been well-studied by researchers from Idenix, Gilead, Pharmasset, Merck, etc. and numerous other academic researchers.”
Dr. Schinazi founded Pharmasset and Idenix with others and is well aware of the literature in this area since he was instrumental in the discovery and development of 2’-fluoro-2’-Me-ribouracil nucleoside analogs. To the best of our knowledge, none of the compounds described in our manuscript have ever been reported in the literature.
As mentioned in our conclusion, each of these compounds had to be prepared through lengthy syntheses (19 to 22 steps) which limited the structural diversity and our SAR studies. We therefore focused our work on 4’-modifications (4’-ethynyl, 4’-cyano and 4’-chloromethyl groups) that were previously reported as being tolerated by other viral polymerases.
Yes, this particular monophosphate prodrug approach has been well studied and validated in human studies (Sofosbuvir, TAF). This is exactly why we are applying it to our newly synthesized related nucleosides.
- “I would recommend the authors to add the synthesis and characterization details as a supporting information file and submit this study as a letter or update; either to this issue/journal if the editor approves it or to another journal.”
As suggested, the synthesis and characterization details have been added to a supporting information file.
- “On a side note: The authors mentioned a supporting information file, but I did not find it. Is this a technical error at MDPI?”
A supporting information file containing protocols along with 1H, 13C and 19P-NMR spectra for compounds 9-14, 16, 17, 20, 24 and 25 has been submitted.
Round 2
Reviewer 2 Report
The revised version of the manuscript reads much better now, and the added supporting information file would be useful for people who want the synthetic procedures.